# One-Fifth of Children with Sickle Cell Anemia Show Exercise-Induced Hemoglobin Desaturation: Rate of Perceived Exertion and Role of Blood Rheology

**DOI:** 10.3390/jcm9010133

**Published:** 2020-01-03

**Authors:** Valentine Brousse, Corinne Pondarre, Cecile Arnaud, Annie Kamden, Mariane de Montalembert, Benedicte Boutonnat-Faucher, Hélène Bourdeau, Keyne Charlot, David Grévent, Suzanne Verlhac, Lydie da Costa, Philippe Connes

**Affiliations:** 1Service de Pédiatrie Générale et Maladies infectieuses, Hôpital Necker Enfants Malades, AP-HP, 75015 Paris, Francebenedicte.boutonnat-faucher@aphp.fr (B.B.-F.); 2LABEX GR-Ex, F-75015 Paris, Francepconnes@yahoo.fr (P.C.); 3UMR_S1134, Inserm, Institut National de la transfusion sanguine, 75015 Paris, France; 4Service de Pédiatrie, Centre Intercommunal de Créteil, 94000 Créteil, France; corinne.pondarre@chicreteil.fr (C.P.); cecile.arnaud@chicreteil.fr (C.A.);; 5Service d’Hématologie Biologique, Hôpital Robert Debré, AP-HP, 75019 Paris, France; helene.bourdeau@aphp.fr; 6Unité de Physiologie des Exercices et Activités en Conditions Extrêmes, Département Environnements Opérationnels Institut de Recherche Biomédicale des Armées, 91220 Brétigny-sur-Orge, France; keynecharlot@gmail.com; 7Service d’Imagerie Pédiatrique, Hôpital Necker Enfants Malades, AP-HP, F-75015 Paris, France; david.grevent@aphp.fr; 8Service de Radiologie, Centre Intercommunal de Créteil, 94000 Créteil, France; suzanne.verlhac@aphp.fr; 9Paris University, F-75010 Paris, France; 10Equipe «Biologie Vasculaire et du Globule Rouge», Laboratoire LIBM EA7424, Université Claude Bernard Lyon 1, 69100 Villeurbanne, France; 11Institut Universitaire de France, 75015 Paris, France

**Keywords:** sickle cell disease, hypoxemia, six-minute walk test, children

## Abstract

Perceived exertion is an important self-limiting factor influencing functional capacity in patients with sickle cell anemia (SCA). Exercise-related hemoglobin desaturation (EHD) may occur during a six-minute walking test (6MWT) and could influence the perceived rate of exertion. The aims of this study were (1) to compare the 6MWT responses (heart rate, perceived rate of exertion, and distance covered) between SCA children with and without EHD, and (2) to test the associations between EHD and several biological/physiological parameters. Nine of 51 SCA children (18%) at steady state (mean age 11.9 ± 3.8 years) exhibited EHD at the end of the 6MWT. The rate of perceived exertion increased with exercise in the two groups, but reached higher values in the EHD group. Heart rate and performance during the 6MWT did not differ between the two groups. The magnitude of change in SpO2 during the 6MWT was independently associated with the red blood cell (RBC) deformability and RBC aggregates strength. This study demonstrates that SCA children with EHD during a 6MWT have a higher rate of perceived exertion than non-EHD children despite a similar physiological demand, and that abnormal RBC rheology determinants appear to be significant contributors.

## 1. Introduction

Sickle cell anemia (SCA) is caused by a single mutation in the beta-globin gene responsible for the production of abnormal hemoglobin S (HbS). HbS can polymerize when deoxygenated, leading red blood cells (RBCs) to sickle. Sickle RBCs have a reduced deformability and are more fragile than healthy RBCs, resulting in chronic hemolytic anemia, repeated painful vaso-occlusive crisis, and various chronic organ damage [1,2,3,4]. In addition, abnormal RBC aggregation properties (increased RBC aggregates strength) have been reported in SCA patients, which may impair blood flow in the microcirculation and participate in the onset of vaso-occlusive events [5,6]. Any decrease in RBC deformability has been shown to result in a decrease of tissue oxygenation and blood flow in the microcirculation [7,8]. 

Adults with SCA have a decreased functional capacity as highlighted by the reduction of the distance covered during a six-minute walk test (6MWT) in comparison with healthy individuals [9,10]. It was previously demonstrated that hemoglobin (Hb) and fetal hemoglobin (HbF) levels, as well as RBC deformability, were independent predictors of the 6MWT performance in SCA children [11]. In addition, the 6MWT performance inversely correlates with the severity of the disease.[12] For instance, a recent study reported an independent association between the 6MWT performance and the presence of silent infarcts in children with SCA and no severe cardiopulmonary dysfunction [13]. Anthi et al. [12] previously showed the usefulness of using the 6MWT distance as an index of pulmonary hypertension and cardiopulmonary function in adults with SCA. Finally, Minniti et al. [14] demonstrated that hemoglobin oxygen desaturation during a 6MWT was more frequent in SCA children with pulmonary hypertension compared to those without.

Exercise-related hemoglobin desaturation (EHD) may occur in a significant proportion of children with SCA during a 6MWT [11,14,15]. The pathological impact of EHD on the clinical expression of SCA patients is unknown, but one may suggest that prolonged EHD during a physical effort could trigger acute complications because of the increased risk of hemoglobin S to polymerize under hypoxemic conditions [11]. Previous studies in healthy endurance trained athletes demonstrated that exercise intensity may contribute to the occurrence of EHD [16]. Although of a submaximal intensity, patients have to walk as fast as possible during a 6MWT in order to cover the longest distance. Indeed, the magnitude of the physiological strain during a 6MWT could predispose SCA patients to develop EHD, but this question has not been completely addressed.

The aims of the present study were (1) to compare the 6MWT responses (heart rate, perceived rate of exertion, and distance covered) between SCA children with and without EHD, and (2) to test the associations between EHD and several biological/physiological parameters. 

## 2. Experimental Section

### 2.1. Patients

Fifty-one consecutive SCA children at steady state from the Necker and the Centre Hospitalier Interrégional de Creteil (CHIC) hospitals were included in this study. The main study (ClinicalTrials.gov: NCT 02909283) was designed to analyze at steady state the biological and physiological determinants of brain metabolism in children with SCA with no vasculopathy or history of stroke. The present work is an ancillary study on the same cohort focusing on the 6MWT responses and the biological and physiological determinants of EHD. Steady-state was defined as no blood transfusion and no acute episode (infection, vaso-occlusive crises (VOC), acute chest syndrome (ACS), stroke, priapism, and splenic sequestration) in the three months prior inclusion. Inclusion criteria also included the following: (1) SS or Sβ°-thalassemia genotype and (2) age between 5 and 17 years old. The charts were reviewed to collect all of the hospitalized VOC and ACS events from birth to the time of study. The rates of the VOC and ACS were calculated for each child by dividing the total number of VOC or ACS episodes by the number of patient-years [17,18]. Ongoing treatment by hydroxyurea (HU) was recorded. 

### 2.2. Six-Minute Walk Test and Pulmonary Function Test

A self-paced six-minute walk test (6MWT) was conducted according to the guidelines of the American Thoracic Society (ATS 2002). The percentage of predicted distance was calculated according to the models of Geiger et al. [19], which take into account the age, height, and gender of the patients. The hemoglobin oxygen saturation (SpO2) and heart rate (HR) were obtained by finger pulse oximetry (Mindray iPM10, Shenzhen, China) before and immediately at the end of the 6MWT. Significant EHD was defined according to the previous studies performed in SCA children [11,15,20], with a drop in SpO2 of 3% or more during exercise compared to resting level. The rate of perceived exertion and the presence/absence of dyspnea were assessed at rest and at the end of the 6MWT. A pulmonary function test performed on the same day was classified as obstructive, restrictive, or mixed according to the American Thoracic Society/European Respiratory Society guidelines [21].

### 2.3. Biological Parameters

The following biological parameters were collected: white blood cell (WBC), platelet (PLT), RBC and reticulocyte (RET) counts, mean cell volume (MCV), hemoglobin concentration (Hb), hematocrit (Hct), HbF, and lactate dehydrogenase (LDH). The blood viscosity was measured at a native hematocrit (Brookfield DVII+ cone-plate viscometer, CPE40-spindle, ≈25 °C, 225 s^−1^). The RBC deformability was determined at 37 °C at 30 Pa by ektacytometry (LORRCA MaxSis Osmoscan, RR Mechatronics, Hoorn, The Netherlands), following recommendations [22,23]. The RBC aggregation was determined at 37 °C via syllectometry (i.e., laser backscatter vs. time; LORRCA MaxSis, RR Mechatronics, Hoorn, The Netherlands), after the adjustment of the hematocrit to 40% [22]. The RBC disaggregation threshold (i.e., the minimal shear rate needed to prevent RBC aggregation or to breakdown existing RBC aggregates) was determined using a re-iteration procedure. 

### 2.4. Ethics

The study was conducted in accordance with the Declaration of Helsinki and was approved by the Regional Ethics Committee (CPP Ile de France III). The children and their parents were informed of the purpose and procedures of the study, and gave written consent.

### 2.5. Statistical Analysis

All values were expressed as means ± standard deviation (SD). As the data collected in the present study followed a normal distribution, an unpaired Student’s t-test was used to compare the different parameters between the two subgroups, namely: SCA children with EHD (EHD) vs. those without EHD (non-EHD). A Chi-squared test was used to test the associations between the qualitative parameters. Pearson correlations were conducted between the changes in SpO2 (rest/end of the 6MWT; i.e., ΔSpO2) and the other parameters. Then, all of the parameters that significantly correlated with ΔSpO2 were included as covariates in a multivariate linear regression model. A *p* value of <0.05 was considered significant. Analyses were conducted using SPSS (version 20, IBM SPSS Statistics, Chicago, IL, USA).

## 3. Results

### 3.1. General and Clinical Characteristics, and 6MWT Responses

The general characteristics of the 51 SCA children are shown in Table 1. Nine of 51 SCA children (18%) exhibited EHD at the end of the 6MWT. The two groups (EHD vs. non-EHD) were not different regarding gender, α-thalassemia status, HU therapy, VOC, and ACS rates (Table 2). Likewise, the pulmonary status was not different between the two groups (Table 2). No child was diagnosed with pulmonary hypertension (i.e., had a tricuspid regurgitant jet velocity ≥2.5 m/s). Conversely, the EHD group was significantly older than the non-EHD group. 

At rest, the SpO2 was not different between the two groups (Table 3). The mean decrease of SpO2 at the end of the 6MWT (compared with the resting condition) was around 6% in the EHD, with ΔSpO2 values ranging from 4% to 9%. The HR was not significantly different between the two groups before and at the end of the 6MWT. Very few children declared dyspnea at rest. At the end of the 6MWT, the proportion of children feeling dyspnea was higher in the EHD group compared with the other group. The presence or absence of dyspnea did not depend on the presence/absence of pulmonary syndromes. Expectedly, the rate of perceived exertion increased with exercise in the two groups, and children with EHD reached higher values than the non-EHD group. The raw distance walked during the 6MWT by EHD children was higher than the non-EHD children, but after correction by gender, height, and age [19], the two groups had the same 6MWT performance (Table 3). Age positively correlated with the raw 6MWT distance (*r* = 0.55; *p* < 0.001), but, as expected, the correlation was lost when the 6MWT distance was expressed as a percentage of the theoretical value.

### 3.2. Biological Parameters

Table 4 shows the biological parameters in the two groups. No difference was observed for the WBC and RBC counts, Hb, Hct, MCV, LDH, aspartate aminotransferase (AST), total bilirubin (BIL), RBC aggregation, and blood viscosity levels. As HU treatment may influence RBC rheology, we also compared the MCV, RBC deformability, RBC aggregation, and blood viscosity between children under HU therapy vs. those without, regardless of EHD status. MCV was expectedly greater in children with HU compared with those without (77 ± 8 fl vs. 87 ± 13 fl, respectively, *p* < 0.01), but no difference was observed regarding the blood rheological parameters between the two populations. However, the EHD children had higher RET and PLT counts, and a lower fetal Hb (HbF) level and RBC deformability than non-EHD children. The RBC disaggregation threshold tended to be higher in the EHD group compared with the other group. 

### 3.3. Correlations and Multivariate Regression Model

We looked for associations between ΔSpO2 and other parameters in the whole cohort, as follows: ΔSpO2 was positively correlated with age (*r =* 0.34; *p <* 0.05), RET count (*r =* 0.27; *p <* 0.05), PLT count (*r =* 0.31; *p <* 0.05), RBC disaggregation threshold (*r =* 0.32; *p <* 0.05), and negatively with the HbF level (*r =* -0.34; *p <* 0.05) and RBC deformability (*r =* −0.42; *p <* 0.01). No correlation between ΔSpO2 and MCV or Hb was observed. A multivariate regression model including these six covariates was significant (R2 = 0.42; ddl = 6; *p <* 0.01), with the RBC deformability and RBC disaggregation threshold being independently associated with ΔSpO2 (*p <* 0.05 for both). Although ΔSpO2 was not associated with the percentage of the predicted 6MWT distance (i.e., performance), a second multivariate regression model was performed including the previous six covariates plus the 6MWT distance. This second model was significant (*R*^2^ = 0.61, ddl = 7, *p <* 0.01), with RBC deformability being independently associated with ΔSpO2 (*p <* 0.01). A positive association was observed between the HbF level and RBC deformability (*r =* 0.64; *p <* 0.001).

## 4. Discussion

This study first demonstrates that SCA children with EHD during a 6MWT have a higher rate of perceived exertion than non-EHD children, despite the fact that the heart rate responses and the normalized 6MWT distance are not different between the two groups. In addition, abnormal RBC rheology appears to significantly contribute to EHD in SCA children.

Very few studies have investigated the changes in SpO2 during exercise in SCA patients. The present study reports that 18% of SCA children had EHD during a 6MWT, a proportion higher than the one found by Campbell et al. [20], but lower than the proportion reported by Waltz et al. [11] and Halphen et al. [15]. The highly selected children included in this study (i.e., with no history of cerebral or cervical vasculopathy screened by transcranial Doppler, a criterion that presumably excludes the most severe hemolytic phenotypes) may explain our proportion. Children with EHD were slightly older than children without EHD, but the multivariate analysis excluded a key role of age in the occurrence of EHD. It has been suggested that the magnitude of the physiological strain during the 6MWT could play a role in the occurrence of EHD [11]. Waltz et al. [11] previously demonstrated that children with EHD covered a higher 6MWT distance (absolute distance and distance expressed in percentage of the predicted distance) than non-EHD children. In the present study, the 6MWT distance covered by the EHD group was also higher than non-EHD group, but this difference is explained by the fact that the EHD children were older than the non-EHD children. Indeed, when normalized by age, gender, and anthropometric characteristics [19], the two groups reached the same 6MWT performance (around 70–75% of the predicted distance). Moreover, the heart rate reached by the two groups during the 6MWT was not different. 

Although the physiological demand was comparable between the two groups in our study, the EHD children had a higher rate of perceived exertion at the end of the 6MWT than non-EHD children. Hypoxemia has been demonstrated to increase the perception of effort in humans through its effects on ventilation, muscle blood flow, and cardiac responses [24]. We were not able to further investigate the cardiac, respiratory, and peripheral responses during the 6MWT in this study, but all children from the EHD group declared dyspnea at the end of exercise, compared with 62% in the other group, suggesting a higher ventilation limitation [25] in EHD children. Further studies investigating cardiopulmonary responses, and notably ventilatory efficiency (ventilation-to-oxygen consumption and ventilation-to-carbon dioxide production ratios) during a 6MWT are needed in order to address this question. Unfortunately, it was not possible to analyze the breath-by-breath gas exchanges during this study, but, importantly, neither EHD nor dyspnea were related to the presence of a pulmonary syndrome (i.e., obstructive, restrictive, or mixed syndrome) nor to the rate of acute chest syndrome. 

Blood rheology is altered in SCA, with patients having a reduction in RBC deformability and sticky RBC aggregates compared with the general population [2,26]. A slight reduction in RBC deformability in exercising healthy athletes and horses has been reported to be associated with the occurrence of EHD in these populations [27,28]. The independent associations found between the levels of RBC disaggregation threshold/RBC deformability, and the degree of SpO2 changes during the 6MWT in SCA children in this study support a role for blood rheology in EHD in this disease, and is in agreement with Waltz et al. [11], who previously reported an independent association between RBC rheology and EHD. Both an increased RBC disaggregation threshold and decreased RBC deformability may increase flow resistance, promote arterio-venous shunts, and disturb microcirculation at the entry of the pulmonary capillaries where RBC aggregates need to be fully dispersed so as to negotiate small capillaries in order to promote adequate gas exchange between the lungs and RBCs [29]. RBC deformability was about 20% lower in children with EHD compared with those without EHD. This difference is physiologically relevant, as Baskurt et al. [30] previously demonstrated that a 15% decrease in RBC deformability was able to increase flow resistance by 75% in isolated perfused rat hind limbs. However, the direct impact of impaired RBC deformability and aggregation on pulmonary hemodynamics was not further investigated, and additional studies are needed to better understand the mechanistic link between impaired blood rheology and EHD in SCA.

Although the difference in the proportion of children treated by HU between the EHD and non-EHD groups was not significant, the percentage was slightly higher in the non-EHD group. Through its effects on HbF synthesis, RBC nitrite levels, and the amount of RBC reactive oxygen species, HU increases RBC deformability [31,32]. The higher RBC deformability in the non-EHD group could therefore be attributable to the slightly higher proportion of children under HU therapy in this group. However, a comparison of the blood rheology between SCA children under HU therapy and those without, unexpectedly showed no difference, despite indirect evidence for adequate adherence through a greater MCV in the treated children. [31] This finding is possibly attributable to the cross-sectional design of this study. Indeed, RBC deformability is highly variable across patients [26], whether treated or not. Nevertheless, the positive correlation observed between the HbF level and RBC deformability supports an indirect role of HbF on EHD, regardless of HU treatment. The effects of HU on EHD need to be addressed by longitudinal studies. 

## 5. Conclusions

In conclusion, while the occurrence of EHD does not always depend on the magnitude of the physiological strain during a 6MWT, the extent of blood rheological abnormalities seems to predispose SCA children to develop EHD. In addition, the presence of EHD is associated with a higher exercise limitation in SCA children.

## Figures and Tables

**Table 1 jcm-09-00133-t001:** General characteristics of the whole cohort.

	SCA Children (*n =* 51)
Gender (M/F)	17/34 (33%/67%)
Age (years)	11.9 ± 3.8
HU (%)	60
Alpha-thalassemia (%)	44
VOC rate (%)	0.6 ± 0.6
ACS rate (%)	0.1 ± 0.1
WBC (10^9^/L)	8.81 ± 3.51
Hb (g/dL)	8.8 ± 1.1
Hct (%)	25.1 ± 3.3
MCV (fl)	83 ± 12
RET (10^9^/L)	221 ± 105
PLT (10^9^/L)	352 ± 158
HbF (%)	16.4 ± 10.0
LDH (IU)	470 ± 163
ASAT (IU)	47 ± 14
BIL (μM)	44 ± 32
DBP (mmHg)	69 ± 11
SBP (mmHg)	111 ± 11

HU—hydroxyurea; VOC—vaso-occlusive crisis; ACS—acute chest syndrome; WBC—white blood cells; RBC—red blood cells; Hb—hemoglobin; Hct—hematocrit; MCV—mean cell volume; RET—reticulocytes; PLT—platelets; HbF—fetal hemoglobin; LDH—lactate dehydrogenase; BIL—total bilirubin; DBP—diastolic blood pressure; SBP—systolic blood pressure; SCA—sickle cell anemia.

**Table 2 jcm-09-00133-t002:** Clinical characteristics of the patients according to exercise-related hemoglobin desaturation.

	Non-EHD Group (*n =* 42)	EHD Group (*n =* 9)
Gender (M/F)	13/29	4/5
Age (years)	11.4 ± 3.6	14.9 ± 3.5 *
α-thalassemia (%)	44	44
HU (%)	62	44
VOC rate (%)	0.6 ± 0.6	0.3 ± 0.4
ACS rate (%)	0.1 ± 0.1	0.1 ±0.1
Normal lung function (%)	38.1	33.3
Restrictive syndrome (%)	28.6	33.3
Obstructive syndrome (%)	21.4	22.2
Mixed syndrome (%)	11.9	11.1

Non-EHD—patients without exercise-related hemoglobin desaturation; EHD—patients with exercise-related hemoglobin desaturation; HU—patients under hydroxyurea therapy; VOC rate—rate of vaso-occlusive crises; ACS rate—rate of acute chest syndrome. Statistical difference: * *p* < 0.05.

**Table 3 jcm-09-00133-t003:** Six minutes walking test (6MWT) responses of the patients according to exercise-related hemoglobin desaturation.

	Non-EHD Group (*n =* 42)	EHD Group (*n =* 9)
SpO_2-_R (%)	98.4 ± 1.4	97.3 ± 2.1
SpO_2-_E (%)	97.8 ± 1.8	91.6 ± 2.6 ***
ΔSpO_2_ (%)	0.5 ± 1.3	5.8 ± 1.6 ***
HR-R (bpm)	91 ± 9	84 ± 4
HR-E (bpm)	143 ± 18	150 ± 26
Dyspnea-R (%)	12	0
Dyspnea-E (%)	62	100 *
RPE-R	0.7 ± 1.2	0.3 ± 0.7
RPE-E	2.6 ± 2.4	4.7 ± 1.2 **
6MWT Distance (m)	531 ± 76	593 ± 38 *
6MWT Distance (%)	72 ± 10	76 ± 7

SpO_2_—Hemoglobin oxygen saturation (R is at rest and E is at the end of the 6MWT); ΔSpO_2_—difference between SpO_2_R and SpO2E; HR—heart rate (R is at rest and E is at the end of the 6MWT); RPE—rate of perceived exertion (R is at rest and E is at the end of the 6MWT). Statistical difference: * *p <* 0.05; ** *p <* 0.01; *** *p* < 0.001.

**Table 4 jcm-09-00133-t004:** Biological parameters of the patients according to exercise-related hemoglobin desaturation.

	Non-EHD Group (*n =* 42)	EHD Group (*n =* 9)
WBC (10^9^/L)	8.6 ± 3.8	9.7 ± 1.7
Hb (g/dL)	9.0 ± 1.1	8.3 ± 1.2
Hct (%)	25.4 ± 3.1	23.9 ± 3.8
MCV (fl)	84 ± 12	76 ± 10
RET (10^9^/L)	211 ± 110	265 ± 64 *
PLT (10^9^/L)	323 ± 152	469 ± 128 *
HbF (%)	17.5 ± 10	8.3 ± 5.3 *
LDH (IU)	449 ± 154	570 ± 196
ASAT (IU)	46 ± 13	48 ± 19
BIL (μM)	41 ± 30	58 ± 37
RBC deformability (30 Pa)	0.51 ± 0.08	0.42 ± 0.08 **
RBC aggregation (%)	51 ± 7	50 ± 5
RBC disaggregation threshold (s^−1^)	339 ± 190	456 ± 195 *^p^*^<0.1^
Blood viscosity (cP)	5.48 ± 0.90	5.03 ± 0.82

WBC—white blood cells; RBC—red blood cells; Hb—hemoglobin; Hct—hematocrit; MCV—mean cell volume; RET—reticulocytes; PLT—platelets; HbF—fetal hemoglobin; LDH—lactate dehydrogenase; ASAT—aspartate aminotransferase; BIL—total bilirubin. Statistical difference: * *p <* 0.05; ** *p <* 0.01.

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
