# Peer review of "One-Fifth of Children with Sickle Cell Anemia Show Exercise-Induced Hemoglobin Desaturation: Rate of Perceived Exertion and Role of Blood Rheology"

_jcm, 2020, doi:10.3390/jcm9010133_

Round 1
Reviewer 1 Report
Abstract
The study population is not adequately described in the abstract. Age, but not sex is mentioned, hemoglobin level is not addressed, neither is whether these patients were taking disease modifying therapy. Did any children have baseline lung disease or were these children excluded?
Introduction
The significance of the 6MWT as it has been studied in adults and children deserves more granular attention and should be tied to specific complications of SCD instead of the bland statement that it is associated with disease severity.
Lines 49 - 50: "Patients with SCA..." should specify if this refers to studies in adults versus children. The citation makes it appear that it is adults.
Methods/Results
The purpose of the study and study design do not match (and therefore neither does the data analysis). The subjects were not enrolled based on their EHD status even though the primary purpose was to compare subjects based on this characteristic. This is a study of a convenience sample of 50 children with SCA who were not, apparently, matched on any fundamental characteristics. Therefore, the first, primary analysis should be a description of the entire cohort. A secondary analysis comparing the two groups is possible, but it does not appear that this study was powered or designed to conduct this comparison and therefore any conclusions comparing these two groups must be significantly softened, especially since the age and hemoglobin F levels of the 2 cohorts was different and these would seem to significantly confound the results of interest.
Curiously, the sex of the participants is not shown. It would be interesting to know the TRV on these subjects.
The general characteristics should include sex and hemolgobin; some measure of HU adherence or other indication...dose in mg/kg? MCV? should be provided.I am not sure why the biological parameters presented in Table 3 are not presented as part of the demographics for this population. They are certainly required to make sense of Table 2.
The issue of hydroxyurea must also be addressed in the discussion where significant attention and significance is placed on differences in rheology between groups with and without EHD. Is the difference seen here attributable to differences in HU use? Again, age also confounds interpretation.
Overall the primary results of this study must be treated as a preliminary and limited secondary analysis and neglect of the effect of hydroxyurea on rheology needs attention, especially given the difference in HbF between the two groups.
https://www.ncbi.nlm.nih.gov/pubmed/?term=Hydroxyurea+rheology+sickle
Author Response
"Please see the attachment"

Reviewer 2 Report
Major Comments:
The study seeks to examine two main questions: 1 whether children with sickle cell disease who have hemoglobin desaturation during a 6 minute walk time test have more feelings of exertion and 2 whether laboratory parameters including red blood cell rheology correlate with hemoglobin saturation during the test.
Overall, the study is well described with clear methodology and results.
The manuscript presumes knowledge of red cell dynamics that a general medical practitioner may not have. If this is the target audience, then the manuscript could benefit from a more detailed definition for exercise related hemoglobin desaturation during the introduction, as well as an explanation for and significance of the rheologic red cell parameters studied. The introduction does not even mention these parameters, but they are a major part of the results and discussion sections are mentioned in the title.
Hemolysis is mentioned in the abstract but largely absent from the manuscript. However, the data the authors present suggests that hemolysis could be hugely important in the pathophysiology of the phenomena they describe. They note in the that platelets, and reticulocyte count are different between the two groups and there are non-significant differences in Hb, WBC, and LDH, suggesting there could be a difference between the two groups in rates of hemolysis. This is also supported by the higher HbF level in the non-EHD group as cells with large volumes of HbF will be less likely to hemolyze. It would be interesting to see if other markers of hemolysis such as AST, or indirect bilirubin were also obtained, or if a hemolytic component could be calculated. The differences in platelets, HbF, and reticulocyte count lose significance in the multivariate analysis, suggesting that they were confounders for the effects of RBC deformability and disaggregation threshold on hemoglobin deoxygenation. Some commentary should be made on what is known about the association between hemolysis and red cell rheology and red cell oxygenation. Could nitric oxide levels play a role in this? NO levels are low in patients with high hemolytic components, NO levels will affect vasodilation in the lungs and periphery and there is some data that NO may also be an allosteric regulator of hemoglobin oxygen affinity.
Minor Comments:
Rate of VOC is normalized for age, however VOCs become more frequent as children age into adolescence and adult hood so a 17 year old in the study may have a relatively low rate of VOC lifelong but a high rate in the past year. It would be helpful if VOC number for the past year was also noted.
The statistical analysis used means with standard deviations and T tests. Does this mean that all the results were normal in distribution? If so this should be stated, if not analysis for non-normal data should be used where appropriate.
It would be easier to compare the lung function tests in Table 1 if a percent of the population was listed after the number instead of a number only.
“Likewise, the heart rate reached by the two groups during the 6MWT was not different, illustrating that EHD is relatively frequent in SCA children, even when the effort is submaximal.” How does the two groups reaching similar heart rates illustrate that EHD is frequent? Why do equal heart rates mean submaximal effort?
Minor editing and English language issues present.
Author Response
"Please see the attachment"

Reviewer 3 Report
Excellent study. The title could be better: One-fifth of children with non-severe SCA show RBC changes and Hb desaturation with excercise? Or...
Can they speculate on cause? i.e. do they have any echo data? Any insights, did the altered rheology come first, or the desaturation? How do they propose to treat these children?Author Response
"Please see the attachment"

Round 2
Reviewer 1 Report
This is a very nice report. The discussion should be expanded to include a discussion of the effect of hydroxyurea on red cell rheology. Vivian Sheehan's interesting work with the oxygenscan might tie in well to this piece since the Point of Sickling test she has developed would seem to tie into the issue of rheologic properities of red cells as a function of oxygen content.
I wondered, in reading this, whether if EHD is treated as a continuous rather than dichotomous variable (perhaps with by each percent drop in SpO2) if any correlations with HbF or other markers of hydroxyurea adherence would be identified. As the issue of HU Dosing remains contentious, one wonders if there is a minimum dose or HbF effect required to protect against EHD. In any case, the effect of hydroxyurea and/or HbF content (which unlike HU was statistically significantly different) on these results requires further discussion than is currently provided; this seems a curious ommission.
Another thought is about the difference in age between the EHD and non-EHD groups. This seems significant since the authors identify that ventilatory efficiency may contribute to their findings. Older children with SCD may have lungs that are functionally different from younger children in terms of the kinds of injury they have sustained. Again, should be part of the discussion.
Reviewer 2 Report
This manuscript presents a well done study with novel data.
The edits made by the authors improve both the clarity of the introduction and depth of the discussion.
Minor Issue:
The gender %s do not equal 100.
|
Gender (M/F) |
17/34 |
